# Investigation of the Performance of Ti6Al4V Lattice Structures Designed for Biomedical Implants Using the Finite Element Method

**DOI:** 10.3390/ma15186335

**Published:** 2022-09-13

**Authors:** Rashwan Alkentar, File Máté, Tamás Mankovits

**Affiliations:** 1Doctoral School of Informatics, Faculty of Informatics, University of Debrecen, Kassai u. 26., H-4028 Debrecen, Hungary; 2Department of Mechanical Engineering, Faculty of Engineering, University of Debrecen, Ótemető u. 2-4., H-4028 Debrecen, Hungary

**Keywords:** direct metal laser sintering, finite element method, lattice structure

## Abstract

The development of medical implants is an ongoing process pursued by many studies in the biomedical field. The focus is on enhancing the structure of the implants to improve their biomechanical properties, thus reducing the imperfections for the patient and increasing the lifespan of the prosthesis. The purpose of this study was to investigate the effects of different lattice structures under laboratory conditions and in a numerical manner to choose the best unit cell design, able to generate a structure as close to that of human bone as possible. Four types of unit cell were designed using the ANSYS software and investigated through comparison between the results of laboratory compression tests and those of the finite element simulation. Three samples of each unit cell type were 3D printed, using direct metal laser sintering technology, and tested according to the ISO standards. Ti6Al4V was selected as the material for the samples. Stress–strain characteristics were determined, and the effective Young’s modulus was calculated. Detailed comparative analysis was conducted between the laboratory and the numerical results. The average Young’s modulus values were 11 GPa, 9 GPa, and 8 GPa for the *Octahedral lattice* type, both the *3D lattice infill* type and *the double-pyramid lattice and face diagonals* type, and the *double-pyramid lattice with cross* type, respectively. The deviation between the lab results and the simulated ones was up to 10%. Our results show how each type of unit cell structure is suitable for each specific type of human bone.

## 1. Introduction

Due to the significance of medical implants inserted into the human body, current research aims to create lightweight medical implants with good mechanical and biocompatible properties that suit the patient under treatment. A medical implant plays the role of a new part of the body, which makes investigating the interaction between the implant and the surrounding body very important in mechanical and biomechanical terms. Hip implant replacements are increasing in numbers. Patients each have a specific weight, a specific body type, and a specific way of moving their body and limbs. Consequently, the design of the hip implant varies depending on each patient’s parameters.

This variety of designs dictates the need for more complex structures, which are harder to manufacture using the traditional manufacturing methods. Additive manufacturing (AM) techniques offer a good solution to this problem. This technology enables the fabrication of complicated designs for the structure of the hip implant. AM techniques can manufacture an accurate implant that meets all of the required specifications. The possibility offered by the AM techniques—to be able to design, simulate, and then manufacture, no matter how complex the design—has revolutionized the implant manufacturing process. AM can accommodate design optimization methods such as lattice topology optimization, which makes the implant closer in structure and strength to human bone.

Since AM techniques use layer-by-layer geometry building, they prevent the cost of wasted materials that is usually associated with traditional methods of manufacturing. This feature makes AM perfect for manufacturing implants from expensive alloys, such as Ti alloys [1]. Many AM techniques have been researched in order to choose a good manufacturing technique that suits the properties of titanium alloys. Power bed fusion (PBF) techniques—such as selective laser melting (SLM) electron-beam melting (EBM), and direct metal laser sintering (DMLS)—are preferable with titanium alloys due to their high-quality direct near-net-shape fabrication [2]. In [3], Liu conducted a comparison between three methods of AM manufacturing processes: directed energy deposition (DED), SLM, and EBM. It was proven that the DED and SLM processes can be used to manufacture structures with 100–200 MPa higher ultimate tensile and yield strengths compared to the results of EBM. The deviation in lattice structures manufactured using SLM technology was discussed in Radek’s work, where the research focused on how deviations in manufacturing affected the mechanical properties of the lattice structures. A universal FEA material model applicable to a range of strut diameters was also introduced [4].

The DMLS process uses an Yb (ytterbium)-fiber laser that melts a powdered metal to build a complex structure layer by layer [5]. This is a common process with titanium alloys, and its quality is affected by the exposure parameters, the powder material, the temperature at the building platform, and the inert gas flow [6,7]. Since DMLS, in particular, and the PBF techniques, in general, use the layer-by-layer build in manufacturing, the grains being built tend to have an orientation along the building axis. This algorithm differentiates AM techniques from the other traditional manufacturing methods, and gives the ability to control the material’s distribution and orientation [8].

Hip implant insertion includes replacing the injured hip with an artificial prosthesis to transfer loads from the acetabulum to the femur by inserting a stem into the latter [9]. Traditionally, the stem of the implant should be completely in contact with the surrounding area of the femur bone in order to attain a stable fixture for the prosthesis [10]. Kuiper et al. stated that the most common reason for revision surgery on hip implants was a loose stem (in 64% of cases) [11]. The most common reason for stem loosening is known as stress shielding, which occurs when the hip implant has a much greater modulus than the bone into which it is inserted [12].

Most researchers apply the lattice structure technique, which reduces the stiffness of the stem of the hip implant to an acceptable level to match that of human bone [13]. A lattice structure is an internally architectural geometry that is composed of repeated unit cells. The parameters of the unit cells’ strut diameter/length, geometry, and orientation lead to various lattice geometries with different material properties. Thus, Abdulhadi and Mian tried to adapt these properties to suit the biomedical application area by controlling these parameters [14]. Arabnejad et al. reported that applying a porous structure with Ti6Al4V alloy can improve bone ingrowth by 56% [15]. Pobloth et al. proved, via an animal trial, that the regeneration of Ti alloy scaffolds with low stiffness is better [16]. In search of the strongest lattice structure, Camil investigated six types of lattice cells manufactured using the PolyJet technology. The research determined the strongest structure by calculating the maximal compressive strength [17]. However, these mentioned studies did not resolve the problem of finding the best match between the type of lattice structure and the part of the human bone to be replaced.

Various classifications have been used to designate the lattice structure types, e.g., periodic or random; 2D or 3D; open or closed; homogeneous or heterogeneous [18]. However, the most used type is strut-based [19]. Gürkan tried to manipulate the parameters of three unit cell types: octahedral, star, and dodecahedral [20]. The samples were manufactured from Ti6Al4V powders using the laser PBF technology. The study concluded four main useful results:Low-strut-diameter lattice structures were successfully manufactured using SLM.On the parallel building platform, the surfaces were smoother and better finished compared to the side surfaces.The octahedral lattice structure had the best dimensional accuracy among all types.The dimensions of the 3D-printed structures were smaller than the nominal CAD values, proving the shrinkage of the material upon solidification.

Flávio reviewed the mechanical properties of the Ti6Al4V alloy with laser bed fusion as a manufacturing method. The review stated that Ti6Al4V is a perfect lightweight structural material due to its high specific mechanical properties and excellent corrosion resistance. The research also mentioned that implants made from materials with over 900 MPa yield strength and over 1000 MPa tensile strength would perform better [21]. Jamari stated that evaluating the contact pressure of the bearing material could be a very good factor in choosing raw materials. The study found that using Ti6Al4V is the best option because it can reduce contact pressure by more than 35% compared to other metals [22].

In pursuit of reducing the stress-shielding effect, Yuhao applied lattice optimization using face- and body-centered cubic structures with vertical struts on the stem of the hip implant [23]. The study reported more than 50% less stress shielding and a noticeable increase in the life cycle. In [24], Jorge conducted a CT scan on a patient’s hip, and estimated the density of the bone in the affected area with the help of computer software. The research succeeded in proposing an implant designed based on the anatomy of the patient, yet maintaining the internationally standardized dimensions of the hip implant. The unit cell design chosen was a triply periodical minimal surface (TPMS) with tunable parameters to make implant as close to the required density as possible. Fei Teng discussed how the shape of unit cells has an effect on the mechanical properties of the structure by comparing four types of lattice structure. The study confirmed the curved gyroid structure to have the best mechanical properties among all types [25].

Today, designers try to validate the results of the optimization applied to manufactured implants via laboratory compression tests and simulation using the finite element method. Nathanael manufactured two hip stems using AM, and each of them had a different internal lattice design: a stochastic porous structure, and a selectively hollowed approach [26]. The research measured the stem stiffness and predicted the reduction in stress shielding with a compression test, where both stems were compressed in the same way that forces act on hip stems in real life, i.e., along the same axes. The results showed a reduction in stiffness of 39% and 40% for the porous and hollowed implants, respectively, and a more load-transfer-compliant implant was designed.

Rasoul used compression tests on lattice structures manufactured by the SLM technique [27]. The aim of the study was to examine the effect of the cellular structure on the strength of the manufactured samples. The research concluded that cellular structures can be responsible for reducing compression strength by up to 60%.

Cheng applied compression tests with two kinds of cellular structures: stochastic foam, and reticulated mesh [28]. The samples were manufactured from a titanium alloy using EBM with high applied porosity (up to 62–69%). Through microscopic investigation during the compression stage, the research found that these structures have comparable compressive strength (4–113 MPa) and elastic modulus (0.2–6.3 GPa) to the values of trabecular and cortical bone.

Mahdi investigated cellular-structured hip implants in terms of fatigue using a compression fatigue test [29]. The samples were manufactured using EBM and made of Ti6Al4V. ANSYS software was also used to conduct a finite element analysis. The study concluded that compression tests and the simulated tests could express the effects of surface roughness by using the fatigue strength factor, where the increase in the fatigue factor caused an approximately 100-fold increase in the fatigue life. Zhao studied the deformation behavior and the failure mechanism using compression tests on lattice structures manufactured using SLM [30]. The approach showed the deviation and the agreement between the CAD-based structures and the manufactured ones.

Finite element analysis (FEA) is used to simulate real laboratory experiments to obtain a preliminary insight into the behavior of the implant. However, researchers have stated that a deviation is inevitable between the real-lab tests and the simulated ones. Helou et al. tried validating the cellular structure data derived from FEA and the empirical data of a compression test. The results showed an error in each specimen that described a ratio of the Young’s modulus from FEA to the Young’s modulus found through empirical testing. The conclusion stated that a negative error means that the simulated specimen is stronger than the actual one, whereas a positive error means that the simulated specimen is weaker than the empirical one [31].

Although results might differ between reality and the FEA results, Zheng suggested that there is a sort of agreement in the results. His research studied lattice structures with different shapes and densities using FEA. The findings of his work gave a prediction of the mechanical properties of the lattice structure, where Young’s modulus was increased with the increase in the relative density [32].

Another in vivo loading study on lattice structures confirmed, using FEA, that the optimal design for hip implant geometry in terms of stress behavior was achieved with functionally graded lattice structures. The research stated that hip implants could withstand up to twice the in vivo load [33]. Ghani used FEA with ANSYS software to predict the elastic modulus of SLM-printed samples of lattice structures. The research confirmed that FEA could be used in the investigation of stress distribution in a three-dimensional model of human bone [34]. Gok was able to reduce the maximum values of stress by up to 25.89% using FEA studies on the multi-lattice structural design of hip implants [35]. Kwang-Min tried to provide an optimized design process for the lattice structures by applying linear static finite element analysis, nonlinear finite element analysis, and experimental tests. The study confirmed that the cubic lattice structure with a 3 × 3 × 3 array pattern had the best axial compressive strength properties [36].

This research is dedicated to investigating the performance of lattice structures to be used in biomedical implants. It further seeks to classify the unit cell types based on their effective Young’s modulus in order to find the best implant structure that suits the patient-specific bone to be replaced. Lattice structure samples were manufactured from Ti6Al4V using the DMLS method. Laboratory tests and FEM were used to study the effects of lattice structure geometry on the mechanical properties. The geometry of the unit cells is represented by the shape and size of the beams forming the structure. Each of the four designs had a different shape of accumulated unit cells with a specific beam size. The porosity, however, was kept within the same range in order to present a valid comparison between all types. The four types of unit cells investigated in this study were chosen based on their levels of complexity, starting with the simplest type (3D lattice infill pattern) and finishing with the most complex type (octahedral lattice 2). By investigating the deformation, the displacement along the compression test, and calculating the effective Young’s modulus, the mechanical properties were analyzed and the results were presented.

## 2. Materials and Methods

The research process started with investigating the Ti6Al4V alloy’s properties, and then designing the CAD model of the lattice structures. In this stage, the porosity was calculated for each of the specimens. Once the design was ready, the specimens were printed, measured, and compressed in the laboratory. At the same time, FE analysis was performed on all types. The results were compared and analyzed to reach conclusions.

### 2.1. Raw Material

Titanium alloy and Ti powder (EOS GmbH, Electro Optical Systems, Munich, Germany) were utilized for manufacturing the specimens. The composition of the Ti6Al4V metallic powder used in this research, with an average particle size of 20–80 μm, is shown in Table 1. EOS titanium Ti6Al4V Grade 23 powder is classified as a titanium alloy according to ASTM B348.

The main mechanical properties of the raw material that were previously determined by the authors and used in the finite element simulations are listed the Table 2.

### 2.2. Lattice Structure Design

In order to analyze the effects of the unit cell parameters—i.e., strut thickness and pore size—on the mechanical properties, four types of lattice structures were designed using SpaceClaim within the ANSYS software 2021 R2, with matching dimensions of 20 × 20 × 30 mm. The lattice volume from each design was 20 × 20 × 20 mm, leaving a 5 mm bulk top and bottom to guarantee equal distribution of the compression force over the lattice surface underneath. CAD files were exported in STL format to transfer to the manufacturing stage.

The porosity calculation was carried out using Equation (1):(1)$$\phi \left(\%\right)=\frac{V_{bulk}-V_{structure}}{V_{bulk}}\;\cdot \;100$$
where *V_structure_* is the volume of the latticed structure, and *V_bulk_* is the bulk volume of the specimen. Table 3 shows the parameters of all unit cells. The parameters were chosen based on the idea of getting a close-in-value porosity percentage for all types in order to make sure that the comparison was as effective as possible by having a small range of porosity among the samples.

In Figure 1, the forming unit cell design for each type is shown, with a detailed view of the dimensions of the specimens. ISO standard 13314 was followed in terms of the relationships among the dimensions, where the length, width, and height of each unit cell were equal to or more than 10 times the pore size. In order to achieve closer results to the real behavior of the structures, 5 mm height bulk material parts were designed at both ends of the specimens. This construction of the specimen ensures that the unit cell structure does not come into direct contact with the pressure plate; this avoids the buckling phenomenon, for example.

### 2.3. Manufacturing of Specimens

In this study, the EOS M 290 (Electro Optical Systems, Germany) 3D-printing machine was used to manufacture the specimens; three samples of each type were fabricated. A list of the machine specifics is shown in Table 4.

The machine was provided with a computer using EOSPRINT 2 software to enable the processing of the CAD data. Figure 2 shows the types of manufactured specimens.

A Mitutoyo-type digital caliper (Mitutoyo America Corporation, Aurora, IL, USA) with a precision of ±0.02 was used to measure the dimensions of the printed specimens, and an Ohaus Navigator scale (Ohaus Corporation, Parsippany, NJ, USA) with 0.01 g readability was used to weigh them. Table 5 shows the average dimensions and weights of each specimen.

In order to guarantee equal compression on both the upper and lower surfaces, the specimens were leveled on the Z-axis using a CNC milling machine, removing material from both ends of the bulk, leaving the total height to be 30 mm.

### 2.4. Compression Test

An INSTRON 8801 Servohydraulic Fatigue Testing Machine (Norwood, MA, USA) was used to conduct the compression tests. An INSTRON (INSTRON, Norwood, MA, USA) AVE2 video extensometer with an accuracy of 0.5% was used with the compression test in order to obtain an accurate measurement of the displacement during the experiment. The testing process was controlled via the WaveMatrix version 2 software (Norwood, MA, USA). Two white dots were marked on the 5 mm bulk areas as reference dots for the video extensometer to measure strain. The experiments were conducted according to displacement at a constant rate, set to 1 mm/min. A maximum force of 15 kN was applied to each specimen. No lubrication was used, because a very small amount of sideways movement was expected. Three samples of each unit cell design were tested under compression, for a total of twelve quasi-static experiments performed. The testing was executed at room temperature.

The results were evaluated, and stress–strain graphs were generated to be compared with the FEM analysis results. Young’s modulus was calculated for each type, and then all four types were compared. Figure 3 shows the testing setup.

### 2.5. FEM Simulation

The compression test simulation was performed using the ANSYS 2020 R2 Workbench software (Canonsburg, PA, USA). The simulation tests were executed with the same parameters used in the laboratory tests, with a compression force of 15 kN applied to the upper surface of the structure, and a fixed support on the bottom surface. The mesh settings were tested by the software at the default size of 2 mm. However, deformation convergence was achieved at a 1.5 mm mesh size, with a change of less than 1% for all types. The number of elements was in the range between 500,000 and 700,000. In the numerical analysis, the material constants were used as described in Table 2. The simulation parameters were unified for all types of specimens. Figure 4 shows the simulation settings of loadings and fixation for the specimens.

Stress–strain curves were generated from the FE results. The effective Young’s modulus for each type was calculated using the computed data.

To ensure an accurate outcome of the FEM analysis, a mesh convergence investigation was conducted based on the deformation and the mesh element size. The change in values was very small, meaning that the element size was suitable to produce accurate values. Figure 5 shows the element size and the deformation convergence.

## 3. Results

### 3.1. Compression Tests

The results of the compression tests were evaluated for each set of specimens. Stress–strain curves are shown for each type in Figure 6. Since the compression was considered in the elastic zone, it was clear that the relationship between stress and strain is linear. For all graphs, the results were shown up to 0.004 strain, since the important part was to calculate the effective Young’s modulus. However, as previously mentioned, the maximum loading was up to 15 kN of force.

Figure 7 shows the average results for the four specimens.

The average values of the resulting Young’s modulus show the ability of the lattice structures to reduce the stiffness of the structure by up to 80–90%. The force applied on all of the specimens was of the same value, meaning that the stress was also the same on all specimens. Thus, the change in the effective Young’s modulus value was due to the variation in the strain value, i.e., the reaction of each type against the stress applied. The highest value of the effective Young’s modulus was for the “octahedral lattice 2” type, making this the stiffest geometrical structure among all tested structures, whereas the lowest value was for the “double pyramid lattice with cross” type, which was the least stiff. The other two types showed similar values of stiffness, with reasonable results.

### 3.2. FEM Results

Stress–strain curves were also generated using the results of the FE analysis. Figure 8 shows the results of the effective Young’s modulus for all types of specimens. Curves are also shown up to 0.004 strain for all types. The maximum loading was up to 15 kN of force.

The deviation percentages between the nominal FEM simulation results and the average results of the laboratory compression tests are shown in Table 6.

The maximum deviation in the value of the effective Young’s modulus percentage was up to 10% (in the third type: double-pyramid lattice and face diagonals) between the laboratory-tested specimens and the simulated ones. Many reasons could be behind this deviation—for example, the 3D-printing quality, the material powder quality, and/or the accuracy of the printer.

Since the tensile and fatigue tests require different shapes and geometry of specimens, they were not executed, and were delayed for future work. However, testing the unit cell structures for tension and fatigue would also enhance the results found in this research. Such tests will be conducted on whole implants in future studies.

## 4. Conclusions

This study investigated four types of lattice structures. The linear elastic behavior of each type was studied under laboratory compression tests. The results were then validated using FE analysis. Stress–strain curves were calculated to determine the Young’s modulus values for each test. Based on the values of the lattice structure types, each one is suitable for a specific area of the hip implant. The “Octahedral lattice 2” type showed the highest average Young’s modulus value of 11 GPa, which is close to the properties of the femur bone (under compression) in both genders [39]. The “3D lattice infill pattern” and “Double pyramid lattice and face diagonals” types had an average Young’s modulus value of 9 GPa, which is close to the proximal femur bone’s modulus (under compression) in males [40]. The “Double pyramid lattice with cross” type had the lowest average Young’s modulus value of 8 GPa, which is the closest in value to the femoral head bone (under compression) [41]. There was a maximum deviation of 10% between the simulated and lab tests, attributable to many factors, such as the printing quality, the printing accuracy, and the quality of the material itself.

Applying the lattice structures to the specimens succeeded in reducing their effective Young’s modulus by a high percentage—up to 92% on average. This reduction plays a major role in making the stiffness of the implant structure closer to that of human bone. However, further study is required on each of the types discussed when applied to the whole implant to be designed.

## Figures and Tables

**Figure 1 materials-15-06335-f001:**
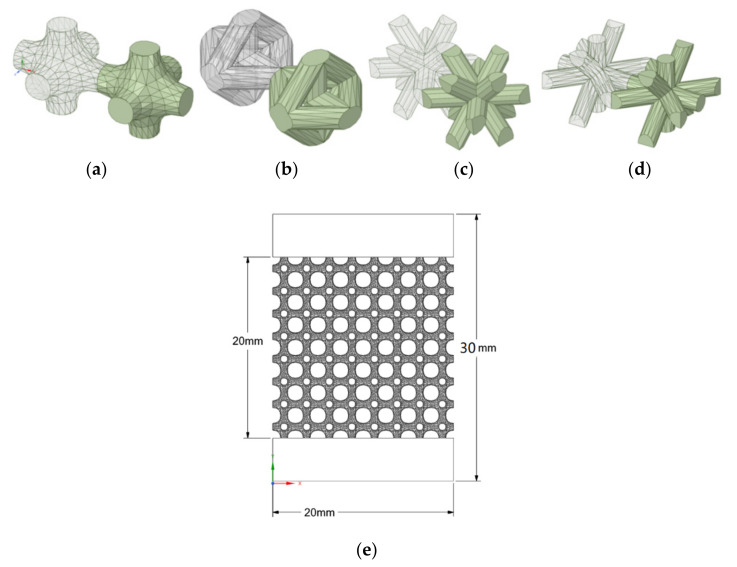
Unit cell design: (**a**) 3D lattice infill pattern, (**b**) double-pyramid lattice with cross, (**c**) double-pyramid lattice and face diagonals, (**d**) octahedral lattice 2, (**e**) specimens’ dimensions.

**Figure 2 materials-15-06335-f002:**
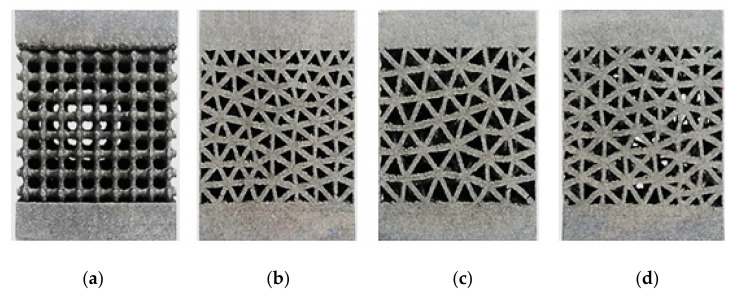
Manufactured specimens: (**a**) 3D lattice infill pattern, (**b**) double-pyramid lattice with cross, (**c**) double-pyramid lattice and face diagonals, (**d**) octahedral lattice 2.

**Figure 3 materials-15-06335-f003:**
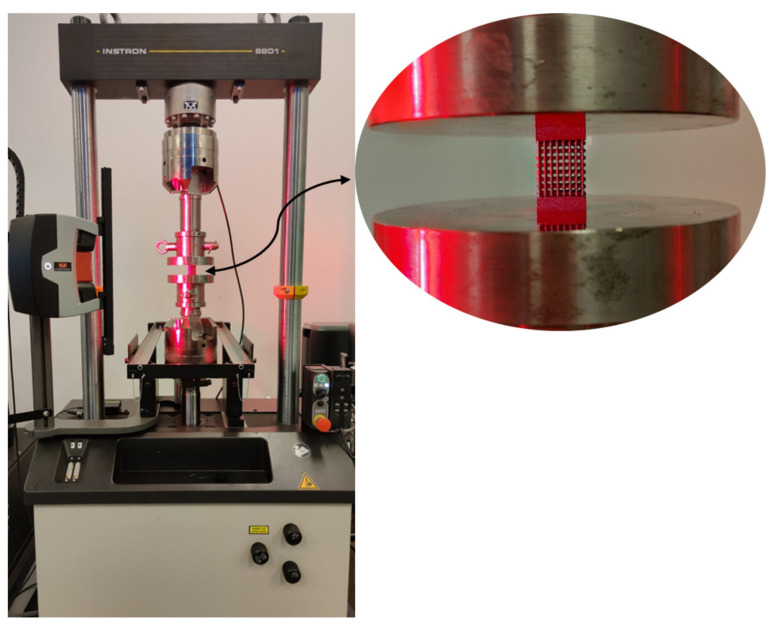
Testing setup.

**Figure 4 materials-15-06335-f004:**
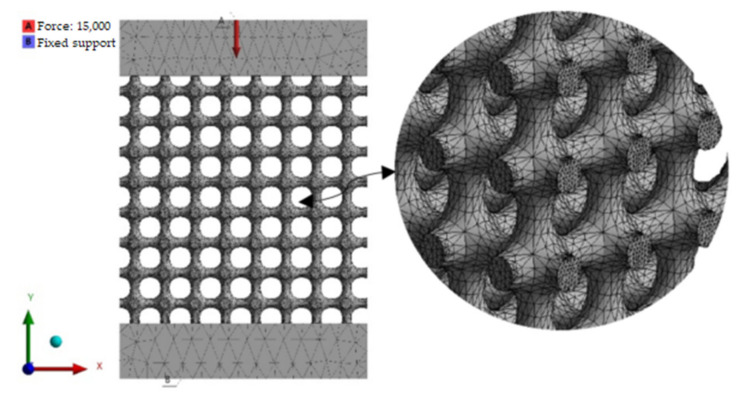
Simulation parameters of the compression test.

**Figure 5 materials-15-06335-f005:**
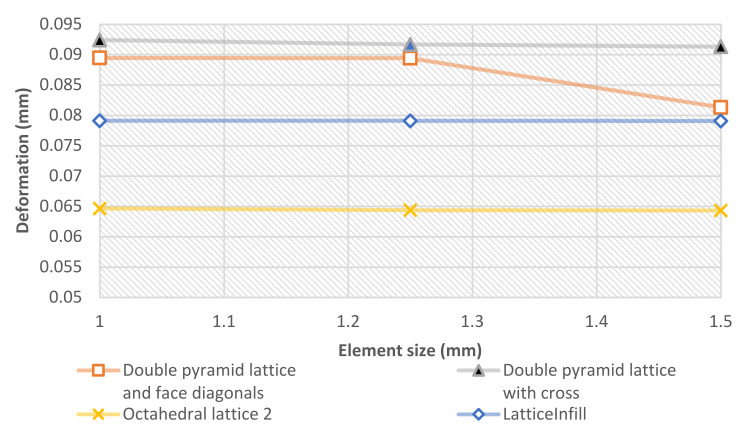
Results of the mesh convergence analysis.

**Figure 6 materials-15-06335-f006:**
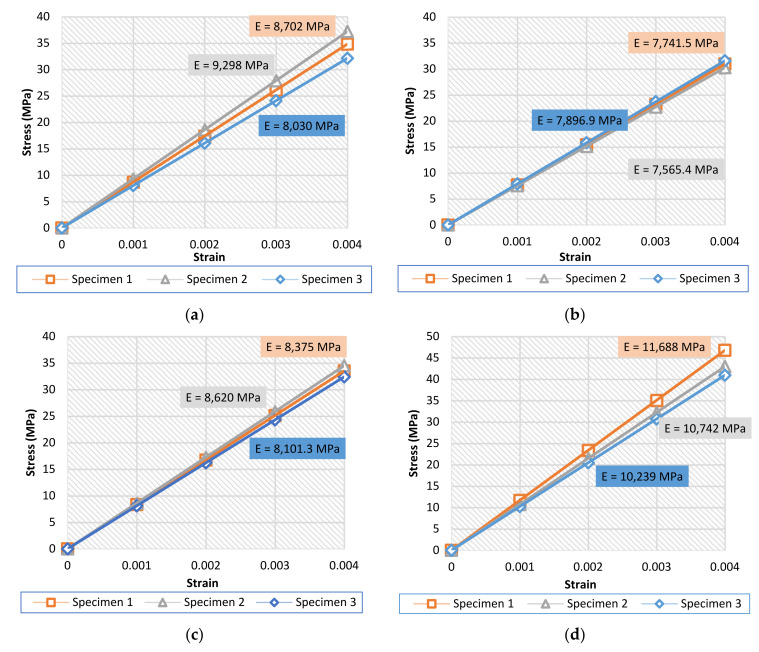
Stress–strain curves with the determined effective Young’s modulus: (**a**) 3D lattice infill pattern, (**b**) double-pyramid lattice with cross, (**c**) double-pyramid lattice and face diagonals, (**d**) octahedral lattice 2.

**Figure 7 materials-15-06335-f007:**
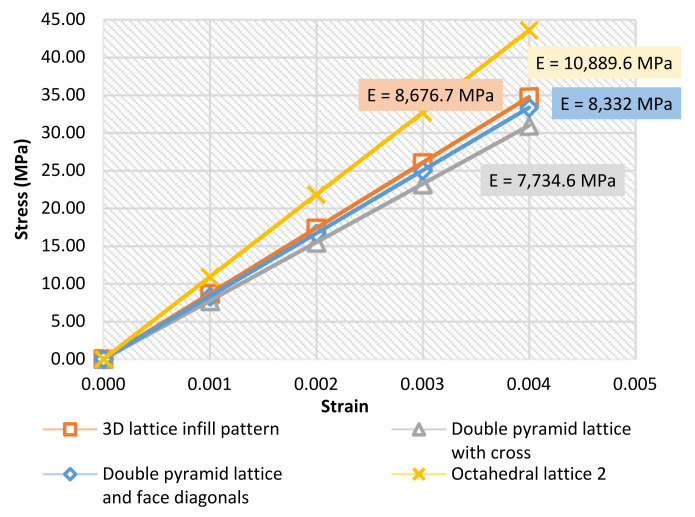
Average stress–strain curves and the average effective Young’s modulus of the specimens.

**Figure 8 materials-15-06335-f008:**
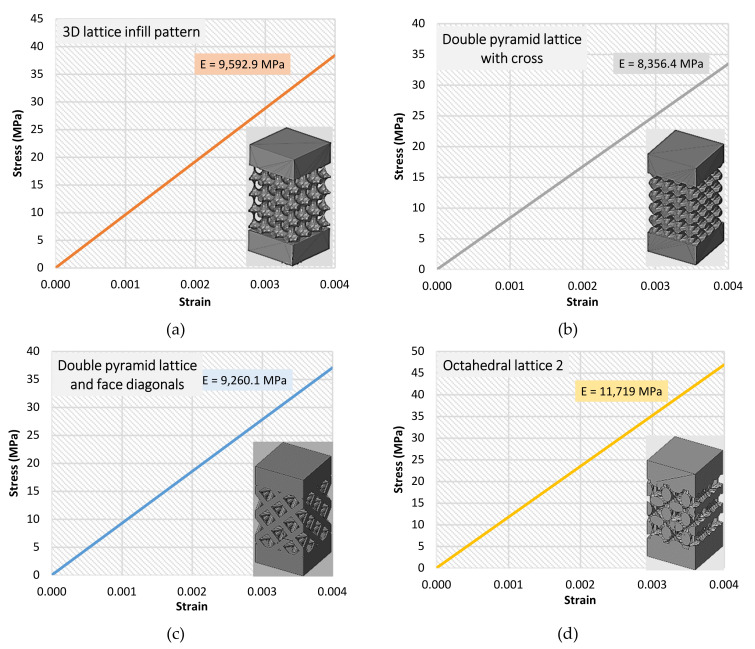
Stress–strain curves and the effective Young’s modulus calculated via numerical analysis. (**a**) 3D lattice infill pattern, (**b**) Double pyramid lattice with cross, (**c**) Double pyramid lattice and face diagonals, (**d**) Octahedral lattice 2.

**Table 1 materials-15-06335-t001:** Chemical composition of Ti64 powder [37].

Element	Chemical Composition Percentage %
Al	5.50–6.50
V	3.50–4.50
O	0.13
N	0.05
C	0.08
H	0.012
Fe	0.25
Y	0.005
Other elements each	0.1
Other elements total	0.4

**Table 2 materials-15-06335-t002:** Material properties used in the FEM.

Property	Value	Unit
Elastic modulus	106,247	MPa
Mass density	4.4	g/cm^3^
Poisson’s ration	0.34	

**Table 3 materials-15-06335-t003:** The parameters of the specimens.

Cell Type	Porosity Percentage (%)	Actual Volume of Latticed Body (mm^3^)	Bulk Volume (mm^3^)	Beam (Strut) Thickness (mm)
3D lattice infill pattern	74	2079.17	8000	0.7
Double-pyramid lattice with cross	74	2067.58	8000	0.85
Double-pyramid lattice and face diagonals	71	2313.96	8000	0.6
Octahedral lattice 2	70	2431.76	8000	0.8

**Table 4 materials-15-06335-t004:** Technical data for the EOS M 290 [38].

Laser Type	Scanning Speed	Focus Diameter	Power Supply	Building Volume
Yb-fiber laser; 400 W	Up to 7.0 m/s (23 ft./s)	100 μm	32 A/400 V	250 × 250 × 325 mm

**Table 5 materials-15-06335-t005:** Dimensions and masses of the specimens.

Unit Cell	Average Length (mm)	Average Width (mm)	Average Height (mm)	Weight (g)
3D lattice infill pattern	19.99	19.97	30.29	26.61
Double-pyramid lattice with cross	20.00	20.00	30.24	26.98
Double-pyramid lattice and face diagonals	19.96	20.03	30.24	28.34
Octahedral lattice 2	20.04	20.03	30.26	28.44

**Table 6 materials-15-06335-t006:** Deviation in the values of the effective Young’s modulus for all specimens.

Cell Type	Effective Young’s Modulus (MPa) (Measurement)	Effective Young’s Modulus (MPa) (Numerical Analysis)	Deviation %
3D lattice infill pattern	8676.67	9592.9	9.6
Double-pyramid lattice with cross	7734.60	8356.4	7.4
Double-pyramid lattice and face diagonals	8332.00	9260.1	10.0
Octahedral lattice 2	10,889.67	11,719	7.1

## Data Availability

All data content can be accessed via this publicly available link: https://mega.nz/folder/fLhmkQ4Q#5-uWL9bxWEs6lk8Ql2GBeg.

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
