# Peer review of "Investigation of the Performance of Ti6Al4V Lattice Structures Designed for Biomedical Implants Using the Finite Element Method"

_materials, 2022, doi:10.3390/ma15186335_

Round 1

Reviewer 1 Report

This paper investigates the unit cell design of four different Ti6Al4V lattice structures for the human bone by testing and simulating the effective elastic modulus, and gives suggestions for the application of human bones for each type of lattice structure. overall, the manuscript is well organized and the results are presented well, and it can be published if the following questions are addressed well.

1.     The novelty shall be included in comparison to previous works.

2.     Why choose 15kN as the applied load for the experiments and simulation?

3.     The mechanical properties of 3d printed materials may vary in different directions. Can you provide a comparison of the testing tensile and compressive modulus of elasticity for the unit cells?

4.     Is it sufficient to evaluate the Ti6Al4V lattice structure for the application of human bones by the only mechanical parameter of effective elastic modulus? Does it need to study other parameters such as fatigue under long-term work, etc.

5.     Conclusion needs to be strengthened, and the outcome of the research should be highlighted.

6.     There are certain typos that needs to be corrected, such as the word “appl.ied” in line 245.

Reviewer 2 Report

1.      The title should be changed to use the MDPI format, changing both the uppercase and lowercase characters.

2.      Including all of the author's email after affiliation with initials if more than one for the same affiliation except for corresponding authors based on MDPI format.

3.      The abstract section must be enhanced with quantitative data.

4.      As your abstract's final sentence, include a "take-home" message.

5.      Please use lowercase font for each of the keywords in accordance with MDPI format.

6.      Please do not use abbreviations in keywords.

7.      The Reviewer do not see the novel in the present article. My examination revealed that several similar previous publications appear to appropriately address the issues you have brought up in the current submission since Titanium alloy lattice structure have been widely discussed in the past. Please emphasize it more advance in the introduction section if there are any more truly something really new.

8.      Previous research has to be explained in the introduction section, including their work, novelty, and limits, to illustrate the research gaps that will be filled in the current study.

9.      Present study shows excellent Ti6Al4V in terms of bio tribological aspect and supported by biocompability aspect from Titanium alloy for medical devices application, especially total hip prosthesis. The introduction and/or discussion part of an article should contain this crucial topic, according to the authors. In addition, to reinforce this explanation, the MDPI-recommended reference should be cite as follows: Jamari, J.; Ammarullah, M. I.; Santoso, G.; Sugiharto, S.; Supriyono, T.; Heide, E. van der. In Silico Contact Pressure of Metal-on-Metal Total Hip Implant with Different Materials Subjected to Gait Loading. Metals (Basel). 2022, 12, 1241. https://doi.org/10.3390/met12081241

10.   Instead of relying just on the predominant text as it already exists, the authors could incorporate an illustration as figures in the materials and methods section that illustrate the workflow of the current study.

11.   It is necessary to provide more information on the manufacturer, country, and specifications of the tools.

12.   The experimental equipment's error and tolerance must be mentioned in the author's work.

13.   An evaluation of the findings with similar past investigations is required.

14.   The present study's limitations should be added before moving on to the conclusion section.

15.   Provide a paragraph-length conclusion rather than the present form's point-by-point description.

16.   In the conclusion section, further research must be discussed.

17.   Five years back literature should be enriched into the reference, and MDPI-published literature is highly recommended.

18.   The authors occasionally created paragraphs in the entire document that were just one or two phrases long, which made the explanation difficult to understand. To make their explanation into a longer, more thorough paragraph, the authors should expand it. It is advised to use at least three sentences in a paragraph, with one serving as the primary sentence and the others as supporting phrases. For example, line 300-3003.

19.   Due to grammatical and language issues, the authors need to proofread the present work. This problem would use MDPI English editing service.

20.   Ensure that the authors followed the MDPI format exactly, edit the current form, and double-check all of the previously noted problems.

Round 2

Reviewer 1 Report

The manuscript can be accepted in present form.

Reviewer 2 Report

The previous comments have been addressed.